# Inhibition of innate immune response ameliorates Zika virus-induced neurogenesis deficit in human neural stem cells

**Pei Xu**[1], **Junling Gao**[1], **Chao Shan**[2], **Tiffany J. Dunn**[1], **Xuping Xie**[2], **Hongjie Xia**[2],
**Jing Zou**[2], **Beatriz H. Thames**[3], **Amulya Sajja**[3], **Yongjia Yu**[4], **Alexander N. Freiberg**[5],
**Nikos Vasilakis**[5,6,7,8], **Pei-Yong Shi**[2,6], **Scott C. Weaver**[3,6,7,8]*, **Ping Wu**[1]*

**1** Department of Neuroscience, Cell Biology and Anatomy, University of Texas Medical Branch, Galveston, Texas, United States of America, **2** Department of Biochemistry and Molecular Biology, University of Texas Medical Branch, Galveston, Texas, United States of America, **3** Department of Microbiology and Immunology, University of Texas Medical Branch, Galveston, Texas, United States of America, **4** Department of Radiology and Oncology, University of Texas Medical Branch, Galveston, Texas, United States of America, **5** Department of Pathology, University of Texas Medical Branch, Galveston, Texas, United States of America, **6** Institute for Human Infections and Immunity, University of Texas Medical Branch, Galveston, Texas, United States of America, **7** Center for Biodefense and Emerging Infectious Diseases, University of Texas Medical Branch, Galveston, Texas, United States of America, **8** World Reference Center for Emerging Viruses and Arboviruses, University of Texas Medical Branch, Galveston, Texas, United States of America

* sweaver@utmb.edu(SCW); piwu@utmb.edu(PW)

**Data Availability Statement:** All relevant data are within the manuscript and its Supporting Information files.

## Abstract

Global Zika virus (ZIKV) outbreaks and their strong link to microcephaly have raised major public health concerns. ZIKV has been reported to affect the innate immune responses in neural stem/progenitor cells (NS/PCs). However, it is unclear how these immune factors affect neurogenesis. In this study, we used Asian-American lineage ZIKV strain PRVABC59 to infect primary human NS/PCs originally derived from fetal brains. We found that ZIKV overactivated key molecules in the innate immune pathways to impair neurogenesis in a cell stage-dependent manner. Inhibiting the overactivated innate immune responses ameliorated ZIKV-induced neurogenesis reduction. This study thus suggests that orchestrating the host innate immune responses in NS/PCs after ZIKV infection could be promising therapeutic approach to attenuate ZIKV-associated neuropathology.

## Author summary

ZIKV has been reported to affect the innate immune responses in neural stem cells. We found that inhibiting the overactivated innate immune responses ameliorated Zika virus-induced neurogenesis reduction in human neural stem cells, which are the origin of the brain. This study suggests that coordinating the host innate immune responses in neural stem cells after ZIKV infection could be promising therapeutic approach to attenuate ZIKV-associated neuropathology.

**Funding:** This work was supported by NIH grants to P.W. (R21AI129509-01), S.C.W. (AI120942), P-Y.S. (AI142759, AI127744, and AI136126). Other funding sources include UTMB CRO Special Fund (P.W.), the John S. Dunn Foundation (P.W., S.C.W., P-Y.S.), the Kleberg Foundation (P-Y.S.), the Amon G. Carter Foundation (P-Y.S.), the Gilson Longenbaugh Foundation (P-Y.S.), the Summerfield Robert Foundation (P-Y.S.), NIH/NIAID 5T35 AI078878 (B.H.T. and A.S.), and McLaughlin Fellowship (P. X.). The funders had no role in study design, data collection and analysis, decision to publish, or preparation of the manuscript.

**Competing interests:** The authors have declared that no competing interests exist.

# Introduction

Zika virus (ZIKV) outbreaks and their strong link to microcephaly and other congenital abnormalities have raised public health concerns globally [1–5]. Clinical and animal studies have shown that miscarriage and brain malformation are more frequent when infection occurs during early pregnancy [1–11]. ZIKV-associated microcephaly occurs most likely due to the high susceptibility of neural stem/progenitor cells (NS/PCs), which populate and develop the fetal brain, to ZIKV infection [12–15]. The exceptional vulnerability of NS/PCs to ZIKV is putatively caused by the highly enriched neurodevelopmental protein Musashi-1 (MSI1) in the neural precursors [16–18], since MSI1 can directly interact with the 3' untranslated region (UTR) of ZIKV genomic RNA to promote viral replication [16]. Recently, Ferraris et al. [19] have shown that ZIKV induces higher viral production and cytotoxic effects in undifferentiated hNPCs compared to differentiated cells. However, it is not clear how these differential responses to ZIKV affect neurogenesis during the neurodevelopmental process, and what is the underlying mechanism.

The Toll-like-Receptor 3 (TLR3) is critical in the recognition of viral nucleic acids to initiate innate antiviral immune responses, including the generation of type I, type II, and type III interferons (IFNs), to provide protection against viral infection [20–24]. In NS/PCs, TLR3 serves as a negative regulator of NS/PCs proliferation in the developing brain [25]. In human cerebral organoids derived from hESC, ZIKV depletes neural progenitors through TLR3 activation, leading to a shrinkage in organoid size, reminiscent of microcephaly [26]. Using hESC-derived NPCs, Liu et al. [27] have shown that abrogating the acute antiviral immune responses in NPCs could mitigate ZIKV infection-triggered growth arrest. Using human fetal brain-derived NS/PC lines, we have previously reported that ZIKV inhibits neuronal differentiation in a cell strain-dependent manner. This inhibition correlates well with the alteration of gene expression patterns in innate immune pathways and neurogenesis, indicating the critical role of innate immune genes in neuronal deficits that result from ZIKV infection [15]. Nevertheless, it is unknown how these innate immune factors affect neurogenesis during the neurodevelopmental process upon ZIKV infection.

In this study, we used strain PRVABC59, the contemporary outbreak Asian-American ZIKV strain, to infect primary fetal brain-derived hNS/PCs at different cell stages: proliferating, priming, and differentiating. We found that ZIKV overactivated key molecules in the innate immune pathways of hNS/PCs to impair neurogenesis, particularly when infection occurs during their proliferating stage. Inhibiting the overactivated innate immune responses ameliorated neurogenesis reduction. This study thus reveals that the overactivated antiviral innate immune response is detrimental to neuronal differentiation, and modulating the host immune responses appears to be a promising therapeutic strategy to attenuate ZIKV infection-triggered neurogenesis deficits.

# Methods

Zika virus. The infectious cDNA clone of ZIKV Puerto Rico strain PRVABC59 (rPRV) was used to rescue challenge virus according to our previous description [28]. All procedures for handling ZIKV at BSL2 were approved by the Institutional Biosafety Committee.

Cell culture and infection. The K048 and G010 strains of primary hNSCs were derived from the human male fetuses cortex [15]. Cells were propagated and primed *in vitro* as described previously [15,29–31]. After 3 days of proliferation and 4 days of priming, cells were differentiated in B27 medium [1: 50 dilution, 1 part B27 media (#17504044, Thermo Fisher Scientific, Waltham, MA) and 49 parts DMEM/F12 stock medium]. Medium was changed every 3 days for 9 days. Both cell strains were used at passage number 25–30. For ZIKV

infections, cells were treated with ZIKV at a multiplicity of infection (MOI) of 0.1 Vero PFU/cell for 1 hour. The number of viable cells after infection was measure by the CellTiter-Glo Luminescent Cell Viability Assay kit (G9242, Promega, Madison, WI, USA), which quantifies ATP-based metabolically active cells.

Quantitative Reverse Transcription PCR (RT-qPCR). Total RNA was extracted from cells or liquid samples using the RNeasy Mini Kit (Qiagen, Hilden, Germany). ZIKV RNA levels were determined as previously described [11,28]. RNA levels of target genes were determined by the iScript One-Step RT-PCR Kit with SYBR Green (Bio-Rad, Hercules, USA) according to the manufacturer's manual on the LightCycler 480 System (Roche, Basel, Germany). Values of genes products ($2^{-\Delta\Delta CT}$) were normalized to glyceraldehyde-3-phophate dehydrogenase (GAPDH). For details on RT-qPCR primer design, see S1 Table.

Immunofluorescence imaging. Cells were fixed in 4% paraformaldehyde (P6148, Sigma-Aldrich, St. Louis, MO, USA)-PBS at 4˚C for 15 mins and were blocked in Tris Buffered Saline (TBS) (T6664, Sigma-Aldrich) plus 10% normal goat serum (005-000-121, Jackson Immuno Research, West Grove, PA, USA), 0.25% Triton-X-100 (BP151-100, Fisher, Waltham, MA, USA) and 2% bovine serum albumin (BSA) (A-4503, Sigma-Aldrich). Cells were incubated with primary antibodies overnight at 4˚C, followed by incubation with secondary antibodies in 0.25% Triton-X-100/TBS for 3 h at room temperature. Primary antibodies were included: rabbit antibodies against ZIKV E protein (1:200, Ab00230-23.0, Absolute Antibody, Oxford, UK); mouse antibodies anti-Flavivirus Group antigen (1:100, MAB10216, MilliporeSigma, Burlington, MA, USA); mouse antibodies against the neuron-specific class III beta-tubulin (Tuj1, 1:2,000, MMS-435P, Covance, Princeton, NJ, USA); rabbit antibodies against glial fibrillary acidic protein (GFAP) (1:2,000, G9269, MilliporeSigma). Secondary antibodies were goat anti-rabbit IgG (1:1,000, R37116, Invitrogen, Carlsbad, CA, USA) conjugated with Alexa Fluor 488 and goat anti-mouse IgG (1:1,000, A-11004, Invitrogen) with Alexa Fluor 568. DAPI (4',6-diamidino-2-phenylindole) was used to stain nuclei at a concentration of 1:5,000. Images were viewed and captured by a Nikon D-Eclipse C1si inverted confocal microscope with the EZ-C1 software v3.50 (Nikon, Minato City, Tokyo, Japan).

Western blotting analyses. Cells were treated by Cell Lysis Buffer (#9803, Cell Signaling Technology, Danvers, MA, USA) supplemented with 1 mM phenylmethylsulfonyl fluoride (93482, Sigma-Aldrich). Protein concentrations were determined by the Pierce BCA Protein Assay Kit (#23225, Thermo Fisher Scientific). Total protein extracts were electrophoresed through the 4–12% NuPAGE Bis-Tris gel (NP0323PK2, Invitrogen) and transferred onto nitrocellulose membranes (#1620115, Bio-Rad). Following blocking, the membranes were probed with primary antibodies at 4˚C overnight and then with secondary antibodies at room temperature for 1 h. Primary antibodies were included: rabbit antibodies against ZIKV NS1 protein (1:1000, GTX133307, GeneTex, Irvine, CA, USA); phospho-Stat1 (Tyr701) (58D6) rabbit antibodies (1:1000, #9167S, Cell Signaling Technology); phospho-Stat2 (Tyr690) rabbit antibodies (1:1000, #4441S, Cell Signaling Technology); Stat1 p84/p91 (C-136) mouse antibodies (1:100, sc-464, Santa Cruz Biotechnology, Dallas, TX, USA); Stat2 (A-7) mouse antibodies (1:100, sc-1668, Santa Cruz Biotechnology); anti-beta 2 Microglobulin rabbit antibodies (1:4000, ab75853, Abcam, Cambridge, UK). Anti-GAPDH mouse antibodies (1:1,000, ab8245, Abcam) were used as internal controls. Second antibodies were horseradish peroxidase-conjugated including goat anti-mouse IgG (1:2,000, 1030–05, SouthernBiotech, Birmingham, AL, USA) and goat anti-rabbit IgG (1:2,000, 4050–05, SouthernBiotech). The antibody-protein complexes were detected by Amersham ECL Western blotting detection reagents (RPN2209, GE Healthcare, Chicago, IL, USA). Chemiluminescent detection of immunoreactive signals was done by LI-COR (Model: 2800, LI-COR Biosciences, Lincoln, NE, USA) and subjected to densitometry analyses using Image Studio Ver 3.1 software.

Bio-Plex assay. Culture medium collected from control and ZIKV-infected cells were used for Bio-Plex assay analysis (Bio-Plex Pro Human Cytokine and Chemokines Assays, Bio-Rad) according to manufacturer's protocols [32]. The assays were run on a Bio-Plex 200 system (Bio-Rad), and the data were analyzed by Bio-Plex Manager (Bio-Rad).

Plaque Assay. Viral titers in the culture medium were determined as previously described [11,33]. Visible plaques in each well of the 24-well plates were counted to calculate the viral titers (PFU/mL).

Statistical Analysis. Cell phenotypic counts using various neuronal and glial markers were conducted on 7–15 randomly chosen areas (~39,000 $\mu m^2$/area) or 350–950 cells per sample, and three to four biological replicates per treatment. All data were analyzed by GraphPad Prism 6 software and presented as the mean ± SD. Data were analyzed by one-way ANOVA with a Tukey post hoc test, or two-way ANOVA with a Dunnett's multiple comparisons test, or $t$-test. A $p$ value of $<0.05$ was considered statistically significant.

## Results

ZIKV replicates rapidly in the proliferating hNS/PCs. In order to study how NS/PCs respond to ZIKV, a human fetal brain-derived NS/PCs culture and differentiation system was established to examine ZIKV infection and its impacts on neurogenesis. The system consists of proliferating and priming as the early stage, and differentiating as the late stage to mimic the neurodevelopmental process (Fig 1A). We have several clinically relevant hNS/PC lines (from individual fetuses K048, K054, and G010), which are all derived from the first trimester human tissues. They are grown as neurospheres and can be primed and differentiated into neurons (Tuj1-labeled) and astrocytes (GFAP-labeled) (Fig 1A and 1C).

In our previous study, it has been shown that both K048 and K054 exhibit similar responses including decreased neuronal differentiation and altered innate immune transcriptomes in response to the Asian-American lineage ZIKV strain Mex1-7, which was isolated from the Mexican outbreak in 2015 [15]. However, G010 was significantly different from K048 and K054, in that it has no ZIKV-induced neuronal reduction and innate immune genes upregulation [15]. Beginning in 2016, Puerto Rico witnessed the most negative consequences of ZIKV infection with the first birth of a ZIKV-related microcephaly baby in the U.S. [34]. Little is known about whether different strains of the Asian lineage ZIKV will have different effects on neurogenesis, although several studies have pointed out the ZIKV of the Asian and African lineage differ in their impact on neural development and antiviral immune responses [35–40]. Therefore, in this study, K048 and G010 were chosen to examine if they would behave differently in response to another Asian lineage stain PRVABC59.

To determine how efficiently PRVABC59 replicated in hNS/PCs, cells at the proliferating stage were infected by recombinant PRVABC59 (rPRV) at MOI of 0.1 for 1 hour, and a plaque assay was done to measure the number of infectious viral particles in the culture medium overtime [28]. Results showed that rPRV replicated rapidly in both K048 and G010 lines of hNS/PCs, and peaked from Days 3 to 7 after infection (Fig 1D), indicating ZIKV replicated more efficiently in undifferentiated hNS/PCs.

ZIKV inhibits neurogenesis in hNS/PCs when infected during the early stage. Clinical and animal studies have shown that miscarriage and brain malformation are more frequent when infection occurs during early pregnancy [1–11]. To mimic the different stages of infection by ZIKV during the course of human brain development, hNS/PC lines were infected by ZIKV at early (proliferating and priming) and late (differentiating) stages (Fig 2A). As shown in Fig 2B and 2C, ZIKV significantly impaired neurogenesis (Tuj1-labeled newly generated neurons) of hNS/PC lines K048 and G010 in a stage-dependent manner, especially when cells were infected

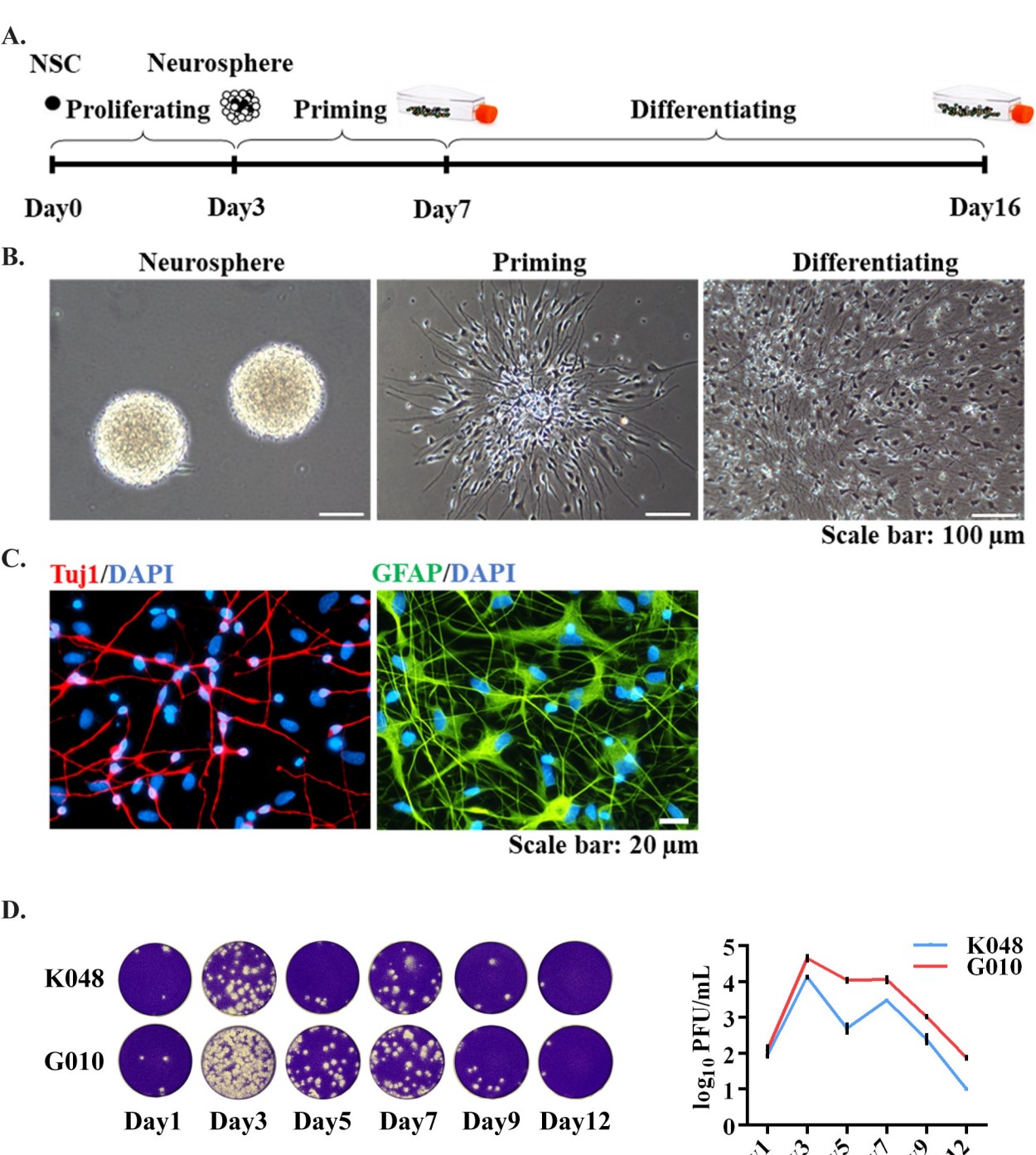

**Fig 1. An *in vitro* hNS/PCs culture system to study ZIKV infection.** (A-C) hNS/PCs are derived from human fetal brains. They proliferate to form neurospheres. After a 4-day priming and 9-day differentiation, they can generate neurons (Tuj1, red) and astrocytes (GFAP, green). Blue, nuclear counterstain. Scale bars: (B) 100 μm, (C) 20 μm. (D) ZIKV replicates rapidly in the proliferating hNS/PC lines. Culture medium was collected on Days 1, 3, 5, 7, 9 and 12 for plaque assay. Representative images of plaques ($10^{-1}$ dilution of the medium) are shown on the left. Data are presented as mean ± SD (n = 2).

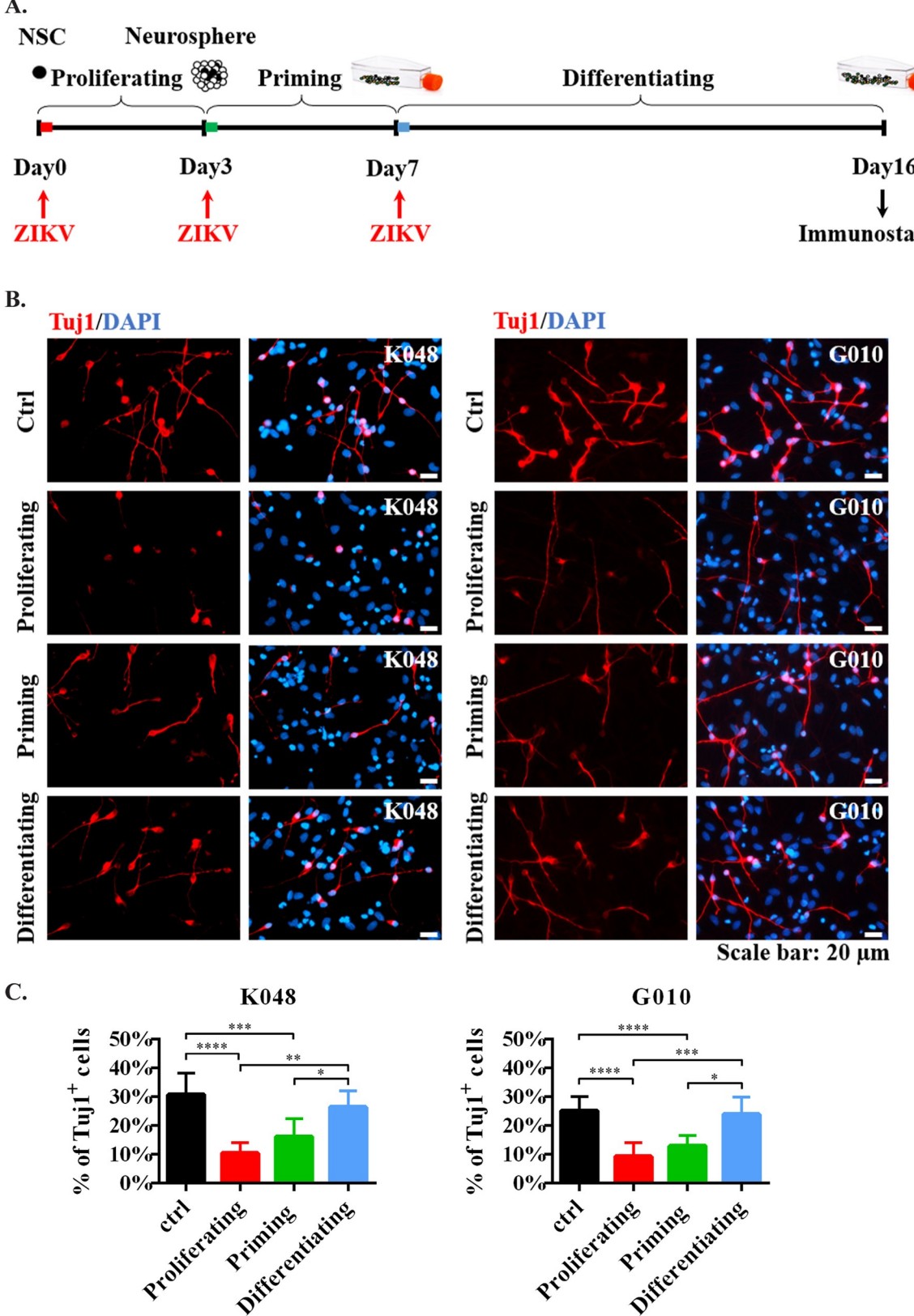

**Fig 2. ZIKV reduced neurogenesis in hNS/PCs during early stage infection.** (A) The *in vitro* system to study ZIKV infection at the different developmental stages of hNS/PCs. (B) Newly generated neurons were stained by Tuj1 (red). Blue, nuclear counterstain. Scale bars: 20 μm. (C) Quantification data are presented as mean ± SD (n = 4), $^*$ $p<0.05$, $^{**}$ $p<0.01$, $^{***}$ $p<0.001$, $^{****}$ $p<0.0001$, one-way ANOVA with a Tukey post-hoc test.

during the early proliferating and priming stages. Similar results were also shown in S1 Fig (newly generated neurons labeled by Tuj1 and MAP2), supporting the stage-dependent effect of ZIKV on neurogenesis. However, our previous study showed no viral-induced neuronal reduction in line G010 upon Mex1-7 ZIKV infection [15]. The differential responses of G010 to Mex1-7 and PRVABC59 suggested that the same host cell may respond to different strains of ZIKV differently.

When looking into the rate of ZIKV infection, only around 1% cells were infected at the end of differentiation (Day 16), and the majority of ZIKV$^+$ cells were GFAP$^+$ astrocytes with very few Tuj1$^+$ neurons (Fig 3A-3C), indicating differentiated astrocytes may be a potential ZIKV reservoir. In fact, using our differentiation protocol, the percentages of newly generated Tuj1$^+$ and GFAP$^+$ cells were around 30% and 45%, respectively (S2 Fig). However, after ZIKV infection, the amount of Tuj1$^+$ cells reduced to 10%, while GFAP$^+$ cells increased to 66% (S4 Fig), indicating that ZIKV promoted astrocytic differentiation but limited neuronal production in neural stem cells [41,42].

ZIKV induces a stronger innate immune response in hNS/PCs infected at the early stage. Several studies have reported that ZIKV infection activates the innate immune responses in NS/PCs [15,26,43,44]. Moreover, we have shown ZIKV-inhibits neurogenesis in a cell-strain-dependent manner, which is well correlated with the innate immune gene transcription pattern [15]. However, it is unclear whether the changes of innate immune responses are also correlated with the infection stages. As shown in Fig 4A, ZIKV was inoculated at the early (proliferating and priming) and late (differentiating) stages to mimic the neurodevelopmental process. The transcription levels of genes in the TLR3/IFN/ major histocompatibility complex class 1 (MHC I) innate immune pathways and neurogenesis were measured one day after the hNS/PCs started to differentiate and the neurogenesis transcription factors began to activate. The results showed that ZIKV robustly enhanced the transcription of most of the tested innate immune genes, including TLR3, interferon regulatory factor (IRF)7, IFR3, IRF1, signal transducer and activator of transcription (STAT)1/2, β2-microglobulin (B2M) and transporters associated with antigen processing 1 (TAP1), especially during early infection in the hNS/PC lines (Fig 4B). B2M is one of the MHC I components, which is a potential pro-aging factor associated with decreased neurogenesis [15,45,46]. Indeed, the transcription factors related to neurogenesis, such as Distal-Less homeobox 5 (DLX5) and doublecortin (DCX), both of which are essential for neuronal migration and differentiation [47,48], were significantly reduced mainly during early infection (Fig 4C), which further confirmed the impaired neurogenesis in hNS/PCs. Additionally, protein expression levels were determined by Western blot to examine how ZIKV affected the IFN signaling in hNS/PCs. Results showed that the ZIKV-induced activation of IFN signaling led to phosphorylation of STAT1 and STAT2, which further upregulated the expression level of B2M (Fig 4D). To determine whether ZIKV alters the cytokines and chemokines production in hNS/PCs, a Bio-Plex assay [32] on the culture media collected from infected cells was conducted. The data (S3 Fig) revealed trends or significant increases of several cytokines and chemokines released from ZIKV-infected cells, including IL-6, interleukin-1 receptor antagonist (IL-1Ra), IL-12, IFNγ, C-X-C motif chemokine ligand 10 (CXCL10), macrophage inflammatory protein-1β (MIP-1β), and regulated upon activation, normal T cell expressed and presumably secreted (RANTES).

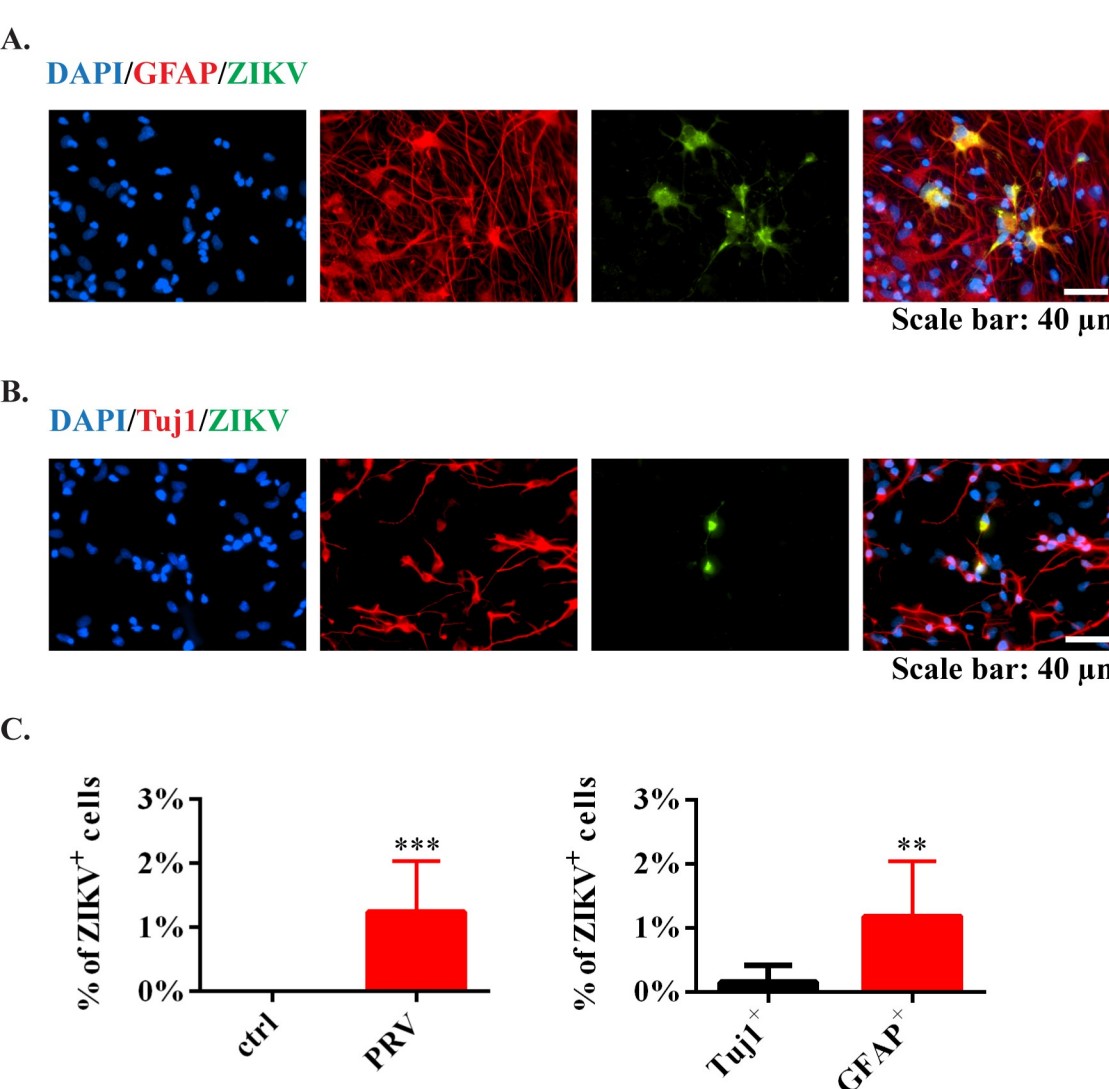

**Fig 3. ZIKV infection in hNS/PC-generated astrocytes and neurons.** (A) Newly generated astrocytes were stained by GFAP (red). ZIKV was stained by anti-ZIKV E protein antibodies (green). Scale bar: 40 μm. (B) Newly generated neurons were stained by Tuj1 (red). ZIKV was stained by anti-ZIKV E protein antibodies (green). Blue, nuclear counterstain. Scale bar: 40 μm. (C) Quantification data were presented as mean ± SD (n = 4), ** $p < 0.01$, *** $p < 0.001$, $t$-test.

Inhibiting ZIKV-induced overactivation of innate immune responses in hNS/PCs ameliorates neurogenesis reduction. The ZIKV-induced neurogenesis deficits were infection stage-dependent, which was well correlated with alternation of innate immune gene expression patterns. Particularly, the transcription level of STAT1 was increased by more than 10-fold after ZIKV infection in the early stage (Fig 4B). STAT1 activation has also been reported to play a critical role in IFNα- and IFNγ-induced NS/PC proliferation and neurogenesis impairments [49–51]. Therefore, we tested a STAT1 inhibitor, Fludarabine (FAMP), in K048 hNS/PCs after ZIKV infection, to see whether inhibiting the overactivated innate immune responses in hNS/PCs could ameliorate neurogenesis reduction. As shown in Fig 5A, FAMP was added to hNS/PCs for two days during proliferation after ZIKV infection. First, we confirmed that FAMP resulted in a specific reduction of STAT1 mRNA but not other STATs (Fig 5B), which was consistent with the literature [52]. Second, we found that FAMP significantly decreased the

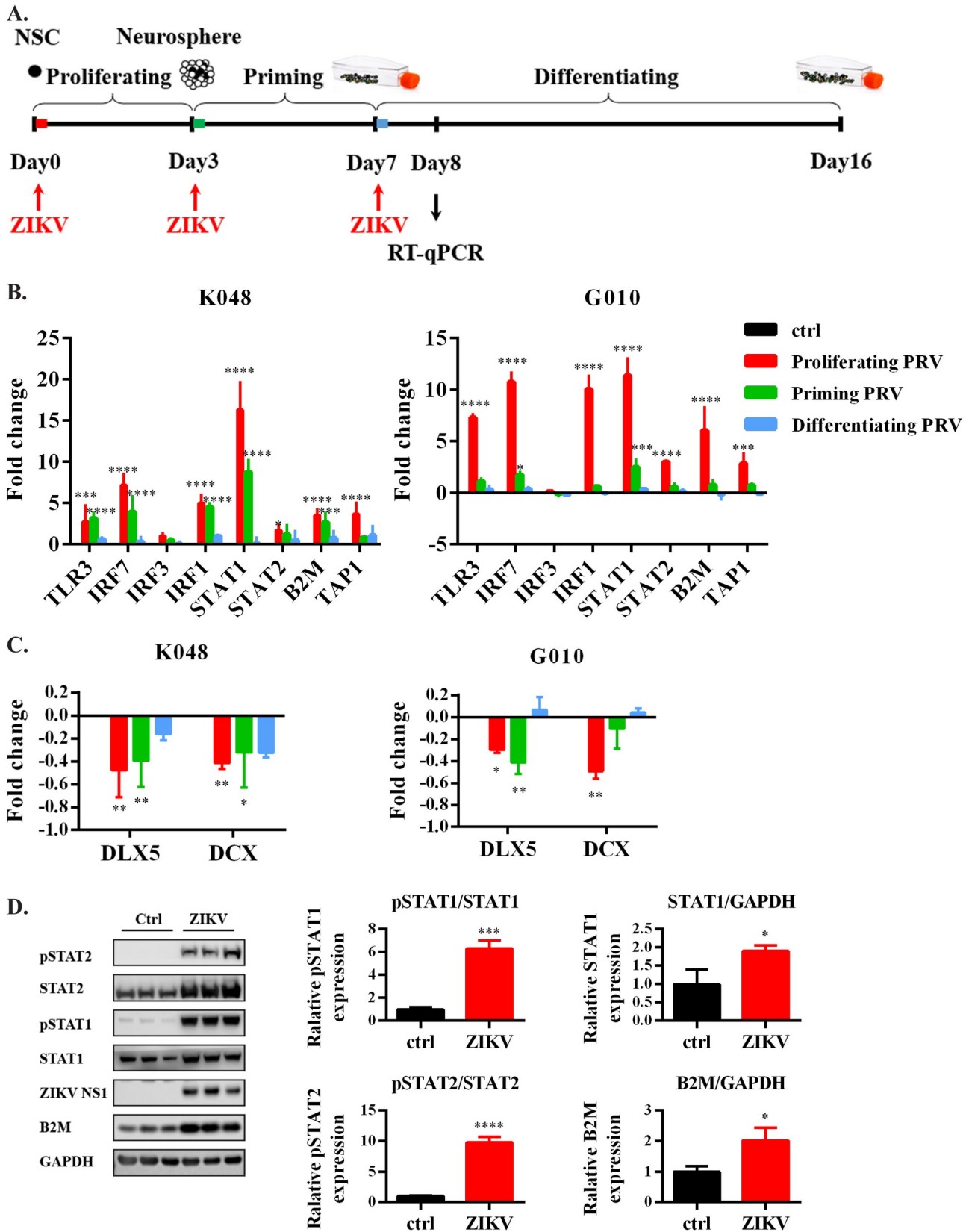

**Fig 4. ZIKV-induced innate immune responses in the hNS/PC lines during the early stage of infection.** (A) The *in vitro* system to study ZIKV infection at different developmental stages in hNS/PCs. (B-C) Transcription levels of innate immune genes were measured by RT-qPCR. Data are presented as mean ± SD (n = 4), * $p<0.05$, ** $p<0.01$, *** $p<0.001$, **** $p<0.0001$, two-way ANOVA with a Dunnett's multiple comparisons test. (D) Protein expression and phosphorylation levels were assessed by Western blot in G010. Data are presented as mean ± SD (n = 3) * $p<0.05$, *** $p<0.001$, **** $p<0.0001$, *t*-test.

ZIKV-mediated neurogenesis impairment, as the percentage of Tuj1[+] cells was significantly higher than the untreated ZIKV (PRV)-infected group (Fig 5C and 5D). Third, the beneficial effects of FAMP on neuronal differentiation after ZIKV infection were not mediated through blocking viral replication in hNS/PCs, as the viral loads in the culture medium from ZIKV-infected cells were similar to those from FAMP-treated, ZIKV-infected cells (Fig 5E). FAMP alone did not affect the cell viability, as the amount of metabolically active cells was not changed with FAMP treatment (S4 Fig). Since FAMP has been reported to have immunosuppressive effects when used to treat hematologic malignancies [52,53], it was more likely that the enhanced neuronal differentiation was due to restricting the ZIKV-induced overactivation of innate immune responses. Interestingly, the transcription level of STAT5a was robustly increased in FAMP-treated ZIKV-infected cells (Fig 5B). STAT5 is expressed in the developing CNS, and involves in neuronal migration and axon guidance during brain development [54,55]. Therefore, the neuroprotective role of FAMP might also be partially caused by the upregulation of STAT5a.

Notably, FAMP ameliorated the neurogenesis deficits in hNS/PCs after ZIKV infection in a stage-dependent manner, as it would not have these protective effects when added on the early differentiation (late) stage. As shown in Fig 6, introducing FAMP at the late stage did not affect the transcription level of STAT1 or the percentage of Tuj1[+] neurons. These data suggested that the intervention is likely to be more effective when applied earlier rather than later.

## Discussion

In this study, we carefully examined the neuropathological effect of ZIKV infection during different stages of neuronal development in a clinically relevant hNS/PC line culture system. The ZIKV-induced neurogenesis deficits were cell stage-dependent, which was well correlated with the altered expression patterns of innate immune genes. The overactivated innate immune responses in hNS/PCs during early stage infection, particularly the robust induction of STAT1 expression and activation led us to investigate the role of STAT1 in neurogenesis impairment after ZIKV infection. We demonstrated that restricting the ZIKV-induced innate immune overactivation by STAT1 inhibitor FAMP ameliorated neurogenesis deficits in hNS/PCs. These data indicate that the overactivated antiviral innate immune response maybe detrimental to neuronal differentiation, and modulating the host immune responses appears to be a promising therapeutic strategy to attenuate ZIKV infection-triggered neurogenesis deficits (Fig 7).

In our hNS/PC culture system, we found that K048 and G010 cell lines responded to ZIKV strain PRVABC59 in a similar pattern, both showing significant upregulation of innate immune responses (transcription factors and genes in the TLR3/IFN/MHC-I pathways) and reduced neuronal differentiation (Tuj1[+] or MAP2[+] cells) during early stage infection (Figs 2 and S1). These results were different from our previous report that K048 and G010 exhibited differential responses to ZIKV strain Mex1-7 [15]. The discrepancy was most likely due to the different virus stains applied to the cells. Therefore, not only different lineages of ZIKV (Asian and African) exhibits different impact on neural development and antiviral immune responses [35–40], different strains from the same ZIKV lineage may also differ in their effects on the same host cell. Combined with our previous findings, ZIKV-induced neuropathology in hNS/

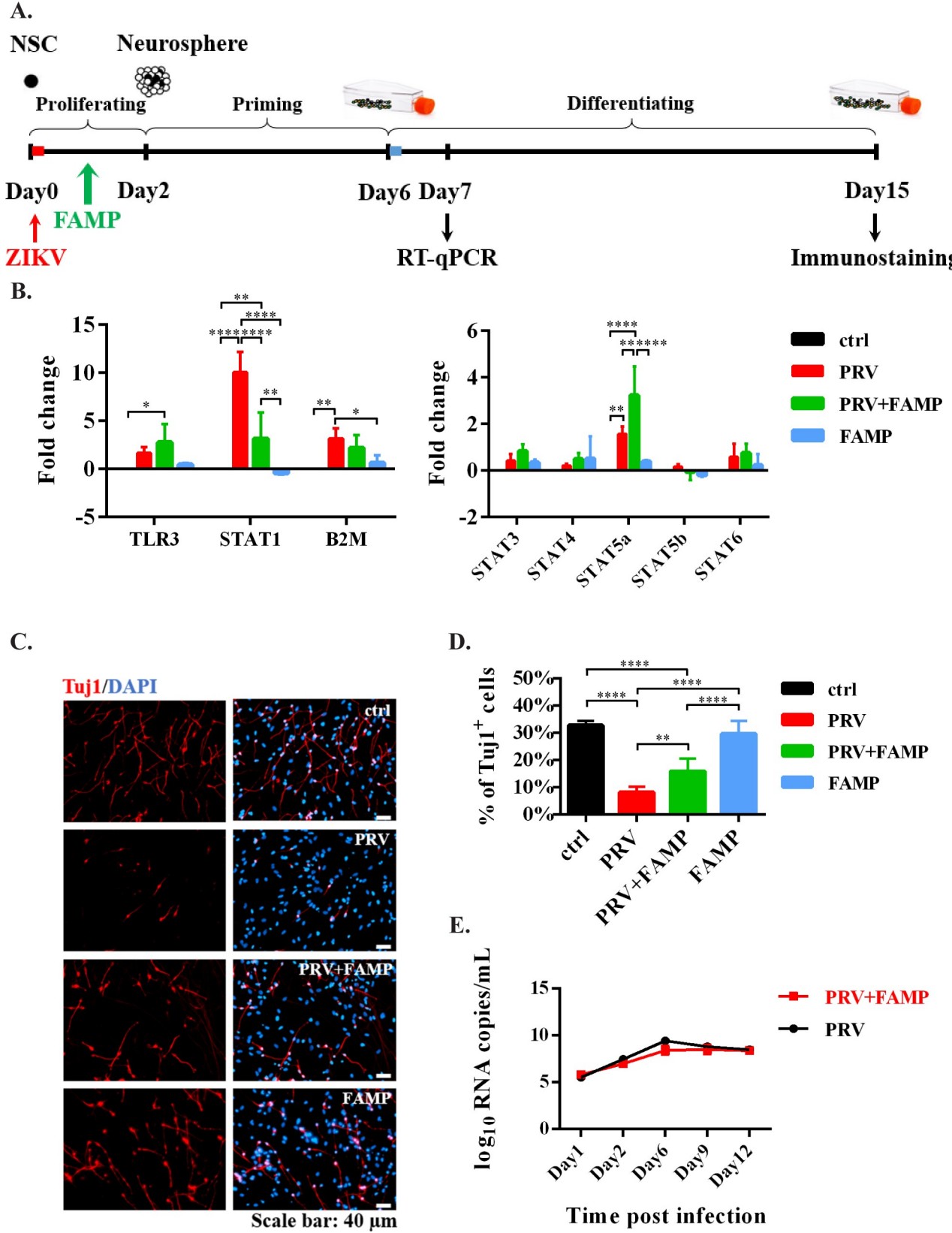

**Fig 5. FAMP-reduced neurogenesis deficits caused by ZIKV infection.** (A) The *in vitro* system to study the effects of FAMP on K048? hNS/PCs after ZIKV infection. (B) Gene transcription levels were measured by RT-qPCR. Data are presented as mean ± SD (n = 4), * $p<0.05$, ** $p<0.01$, **** $p<0.0001$, two-way ANOVA with a Dunnett's multiple comparisons test. (C) Newly generated neurons were stained by Tuj1 (red). Blue, nuclear counterstain. Scale bars: 40 μm. (D) Quantification data are presented as mean ± SD (n = 4), ** $p<0.01$, **** $p<0.0001$, one-way ANOVA with a Tukey post hoc test. (E) ZIKV RNA loads in the culture medium collected on Days 1, 2, 6, 9 and 12 were measured by RT-qPCR. Data are presented as mean ± SD (n = 3).

PCs is both viral strain- and host cell-dependent. The severity of altered neurogenesis in hNS/PCs is determined by the extent of innate immune activation in the host cells [15].

The activation of IFN-associated responses by ZIKV in our hNS/PC lines was similar to the findings by others [15,26,43]. High levels of mRNA of TLR3 and type I/II IFNs were also detected in the peripheral blood of patients with acute ZIKV infection (up to 5 days after the onset of signs/symptoms) [44]. The induction of type I/II IFNs further activated the anti-viral gene expression program by signaling through their receptors type I IFN receptor (IFNAR) and type II IFN receptor (IFNGR) to control viral infection and modulate the antiviral immune responses [56–59]. Particularly, we found that ZIKV-mediated TLR3 activation led to increased protein expression and phosphorylation of both STAT1 and STAT2 in hNS/PCs. This is different from the previous reports that ZIKV inhibits type I IFNs production and downstream signaling in several cell lines, including A549 cells, HEK293 cells, Vero cells and primary human fibroblasts [60–64]. Specifically, the nonstructural protein NS5 of ZIKV antagonizes type I IFN production by reducing the STAT2 expression and by impeding the phosphorylation of STAT1, TANK-binding kinase 1 (TBK1) and IFN regulatory factor 3 (IRF3) [60–64]. The discrepancy was most likely due to the different host cell types. While innate immunity is crucial to defend against microbial invasion, overactivation could be detrimental. Thus, appropriate suppression of the immune response may help to generate a balanced defending mechanism that counters the viral infection while mitigating the tissue damage [56]. In our ZIKV infected hNS/PC lines, the significant neurogenesis reduction was well correlated with the overactivated innate immune responses, although the infection rate of differentiated NS/PCs was less than 2%. These findings suggested that the ZIKV-mediated bystander effects might play a more important role to interfere neuronal differentiation of hNS/PCs.

As a hub transcription factor in the TLR3/IFN/MHC-I pathways, STAT1 controls the expression of numerous downstream factors that influence NS/PCs proliferation and neurogenesis [49–51]. In our study, the transcription level of STAT1 was robustly increased by more than 10-fold after ZIKV infection, while others only altered moderately or mildly (Fig 4B). The STAT1 inhibitor FAMP was then tested in this study to limit the pathological immune responses that could lead to unnecessary collateral neuronal damage. FAMP is a chemotherapy medication that can induce sustained immunosuppression when used to treat leukemia and lymphomas [52,65]. The immunosuppressive effects of FAMP is associated with a selective inhibition of STAT1 [52]. In human peripheral blood mononuclear cells (PBMCs), FAMP specifically depletes STAT1 protein (and mRNA) and blocks STAT1 activation without affecting other STATs [52]. In human brain microvascular endothelial cells (HBMECs), FAMP has been shown to block human immunodeficiency virus type 1 (HIV-1)-induced IL-6 expression and HIV-1/IL-6-induced monocyte migration across the blood-brain barrier (BBB) to attenuate HIV-1-induced BBB compromise, which is relevant to the neuropathogenesis of acquired immunodeficiency syndrome (AIDS) [66].

For the first time we found that FAMP ameliorated the neurogenesis deficits in hNS/PCs after ZIKV infection. Moreover, this protective effect also exhibited a cell stage-dependent manner, as it did not rescue the neuronal damage when applied to the late stage. The

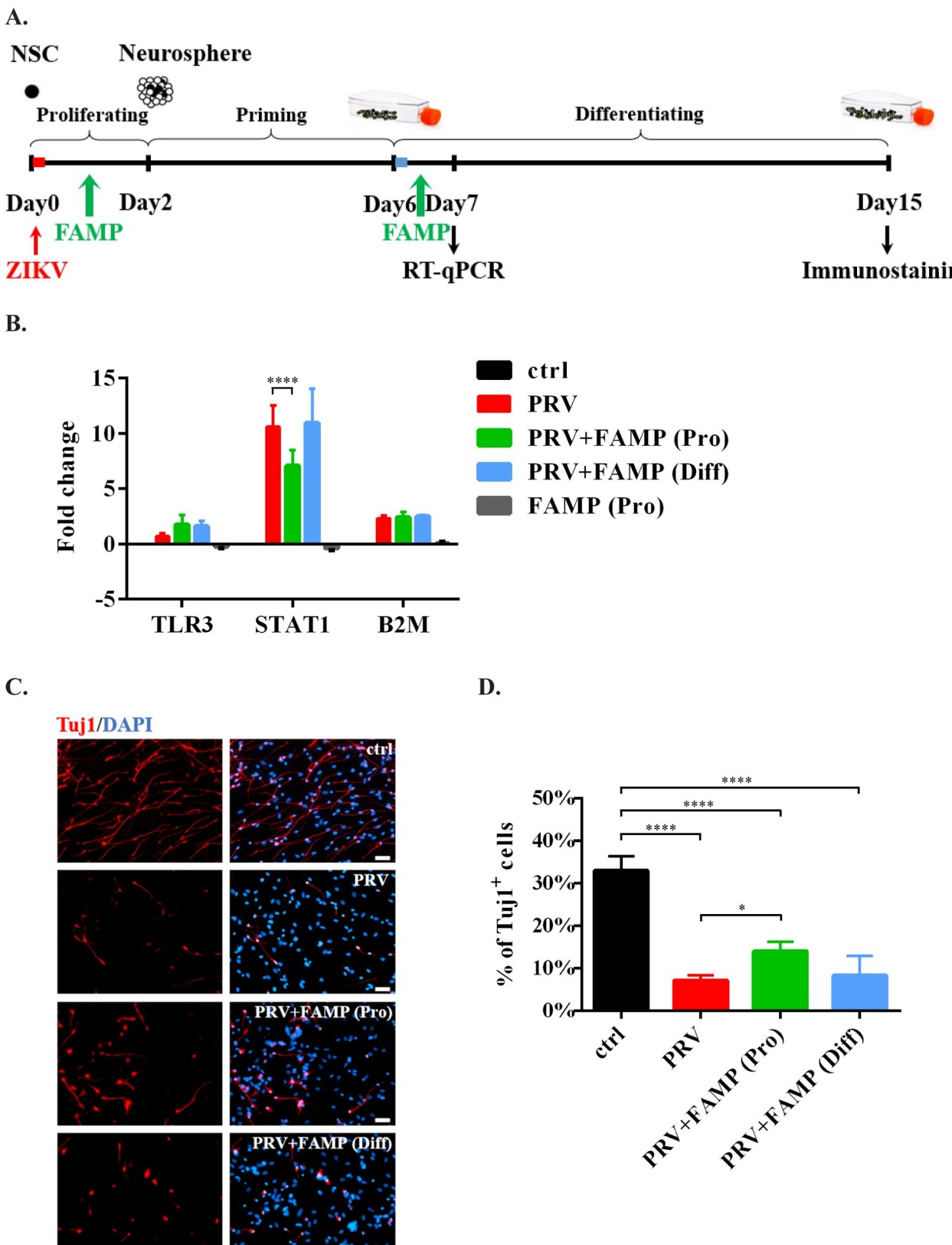

**Fig 6. Lack of rescue of ZIKV-induced neurogenesis deficits with FAMP treatment at the late stage.** (A) The *in vitro* system to study the effects of FAMP on hNS/PCs after ZIKV infection. ZIKV was inoculated at the early proliferating stage. FAMP was treated at the early proliferating or the late differentiating stage. (B) Gene transcription levels were measured by RT-qPCR. Data are presented as mean ± SD (n = 2), **** $p<0.0001$, two-way ANOVA with a Dunnett's multiple comparisons test. (C) Newly generated neurons were stained by Tuj1 (red). Blue, nuclear counterstain. Scale bar: 40 μm. (D) Quantification data are presented as mean ± SD (n = 2), * $p<0.05$, **** $p<0.0001$, one-way ANOVA with a Tukey post-hoc test.

attenuation of neurogenesis impairment by FAMP was not through limiting the viral replication in hNS/PCs, as the viral loads in the culture medium from FAMP treated ZIKV-inoculated cells were close to those from ZIKV-inoculated cells. FAMP did not affect the cell viability either (S2 Fig). More likely, it rescued the neuronal impairment via inhibiting the ZIKV-induced overactivation of innate immune responses in hNS/PCs. Interestingly, the transcription level of STAT5a was significantly increased in FAMP-treated ZIKV-infected cells.

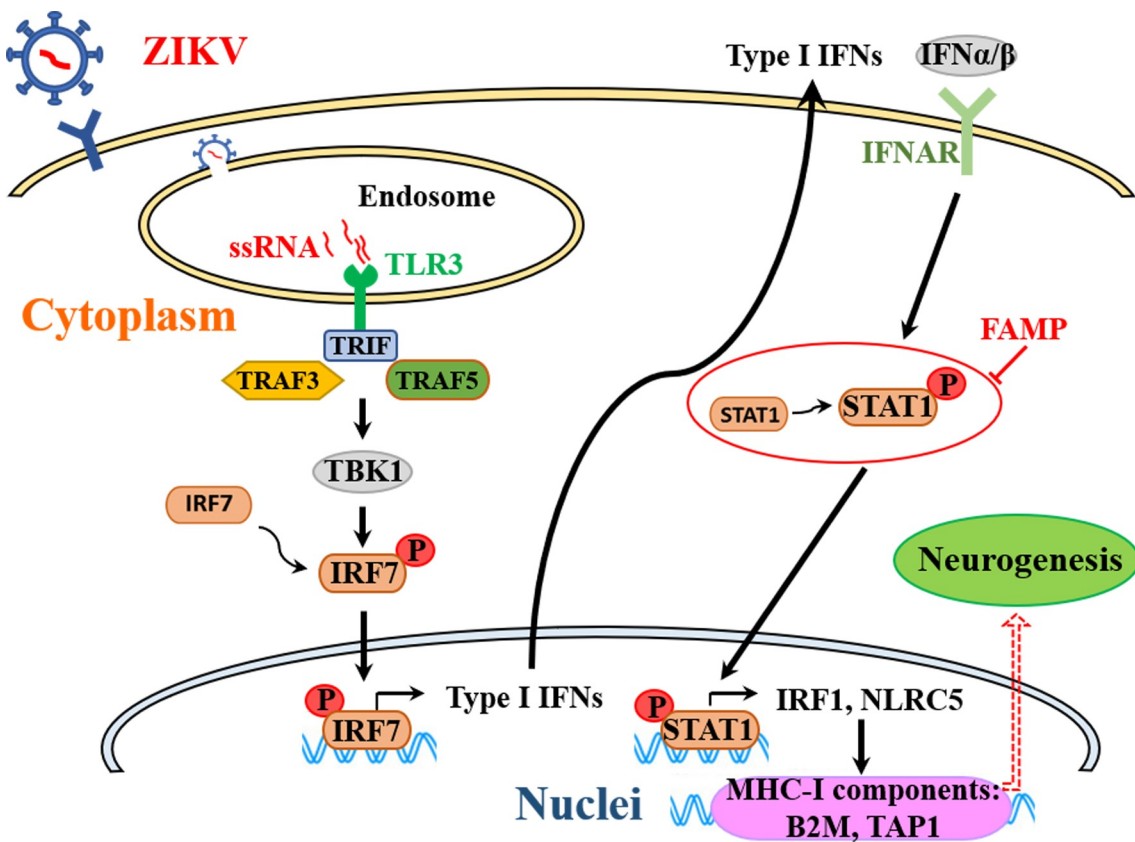

**Fig 7. A hypothetical model of modulating the ZIKV-induced overactivation of innate immune responses in NS/PCs to attenuate neurogenesis deficits.** The entry of ZIKV is mediated by unknown cell surface receptors. The virus membrane then fuses with the endosomal membrane and the ssRNA genome of the virus is released. TLR3 recognizes nucleic acids that are inside the endosome. TRIF is the sole adapter of TLR3. TRIF recruits a signaling complex that is consists of TRAF3 and 5 and the downstream mediator TBK-1. Activation of TBK-1 phosphorylates and activates the transcription factor IRF7 to drive a robust transcriptional activation of the type-I IFN responses. IFN-α and IFN-β bind to the IFNAR to activate STAT1/2 heterodimers. Phosphorylation and translocation of STAT1 leads to activation of transcriptional activity in several STAT1-associated genes, including IRF1 and NLRC5. Both of them can act as transcriptional activators of MHC class I genes, including B2M and TAP1. B2M is a potential pro-aging factor associated with decreased neurogenesis. Targeting the ZIKV-induced overactivated innate immune responses in NS/PCs (e.g., STAT1 inhibitor FAMP) could be a promising therapeutic strategy to attenuate ZIKV-related neuropathology. TRIF, TIR domain containing adaptor protein inducing IFN-β; TRAF, TNF receptor associated factor; TBK-1, TANK Binding Kinase 1; IRF, interferon regulatory transcription factor; IFNAR, IFN-α receptor; STAT, signal transducer and activator of transcription; NLRC5, NLR Family CARD Domain Containing 5; B2M, beta-2-microglobulin; TAP1, Transporter 1, ATP Binding Cassette Subfamily B Member; FAMP, fludarabine.

STAT5 is expressed in the developing CNS, and involves in neuronal migration and axon guidance in the developing brain [54,55]. Therefore, the neuroprotective role of FAMP might also be partially caused by the upregulation of STAT5a. However, the transcription level of STAT5b was not affected by FAMP, possibly due to the structural dissimilarities between STAT5a and STAT5b [67]. Notably, FAMP did not completely reverse the ZIKV-induced neurogenesis deficits, indicating that other ZIKV-mediated bystander effects, such as proinflammatory cytokines and chemokines, may also interfere neuronal differentiation of hNS/PCs [68,69]. Indeed, using Bio-Plex assay on the culture media collected from infected cells, an initial study revealed trends or significant increases of several cytokines and chemokines released from ZIKV-infected hNS/PCs, including IL-6, IFNγ and CXCL10. Further studies are needed to determine their roles in neurogenesis deficits or/and synaptogenesis impairment [70–72].

Among those 2% of ZIKV-infected cells, nearly 90% were GFAP$^+$ astrocytes, which were 8 times more abundant than Tuj1$^+$ neurons, suggesting astrocytes were more permissive to ZIKV infection than neurons [42], and could serve as a potential ZIKV reservoir. Astrocytes are one of the most abundant cell types in the CNS and play critical roles in host defense against viral infections [73–75]. In a developmental immunocompetent mouse model of ZIKV peripheral inoculation in the newborn mouse, it has been shown that ZIKV enters the CNS and initially targets astrocytes throughout the brain, whereas more neurons show ZIKV immunoreactivity at later stages, in part potentially due to ZIKV release from the infected astrocytes [76]. In a mouse model of West Nile virus neuroinvasive disease-induced cognitive dysfunction, it has been reported that the 'preferential' production of proinflammatory astrocytes decrease the adult neuronal differentiation during virus-triggered memory impairment through IL-1 expression [77]. The dual role of infected astrocytes in modulating the innate immune responses to protect the CNS against the viral infection and increasing the release of pro-inflammatory cytokines to reduce neurogenesis needs furthermore characterization.

## Supporting information

**S1 Fig. ZIKV-reduced neurogenesis in hNS/PCs during early stage infection.** (A) Newly generated neurons were stained by Tuj1 (red) and MAP2 (green). Blue, nuclear counterstain. Scale bars: 20 μm. (B) Quantification data are presented as mean ± SD (n = 3), ** $p<0.01$, *** $p<0.001$, **** $p<0.0001$, two-way ANOVA with a Tukey's multiple comparison test. (TIF)

**S2 Fig. ZIKV reduced neurogenesis but incresed astrocytes production in hNS/PCs during early stage infection.** (A) Newly generated neurons were stained by Tuj1 (red), and astrocytes were stained by GFAP (green). Blue, nuclear counterstain. Scale bars: 20 μm. (B) Quantification data are presented as mean ± SD (n = 3), * $p<0.05$, **** $p<0.0001$, one-way ANOVA with a Dunnett's multiple comparisons test. (TIF)

**S3 Fig. Concentrations of cytokines and chemokines in the medium of ZIKV-infected hNS/PCs.** Culture medium were collected from ZIKV infected K048 cells. Concentrations of cytokines and chemokines in the medium were detected by Bio-Plex assay. Quantification data are presented as mean ± SD (n = 3), * $p<0.05$, ** $p<0.01$, **** $p<0.0001$, multiple $t$ tests. (TIF)

**S4 Fig. Viability of hNS/PCs after treatment with ZIKV and FAMP.** Cells were seeded in 96-well plates at a density of 5,000 per well. The viability was evaluated by the CellTiter-Glo Luminescent Cell Viability Assay kit (Promega) according to the manufacturer's instructions. PRV, PRVABC59 strain of Zika virus; FAMP, Fludarabine. Data are presented as mean ± SD

(n = 4), * $p < 0.05$, two-way ANOVA with a Tukey's multiple comparison test.
(TIF)

**S1 Table. Primers used in this study.**
(DOCX)

## Acknowledgments

We thank colleagues (Dr. Jiaren Sun and Dr. Menon Ramkumar) at University of Texas Medical Branch, and Dr. Hongjun Song at University of Pennsylvania for helpful discussions during the course of this study.

## Author Contributions

**Conceptualization:** Junling Gao, Chao Shan, Xuping Xie, Jing Zou, Yongjia Yu, Nikos Vasilakis, Pei-Yong Shi.

**Data curation:** Pei Xu, Junling Gao, Chao Shan, Tiffany J. Dunn, Jing Zou, Beatriz H. Thames, Amulya Sajja.

**Formal analysis:** Pei Xu, Ping Wu.

**Funding acquisition:** Beatriz H. Thames, Amulya Sajja, Pei-Yong Shi, Scott C. Weaver, Ping Wu.

**Investigation:** Pei Xu, Junling Gao, Chao Shan, Tiffany J. Dunn, Jing Zou, Beatriz H. Thames, Amulya Sajja.

**Methodology:** Pei Xu, Junling Gao, Chao Shan, Tiffany J. Dunn, Jing Zou, Beatriz H. Thames, Amulya Sajja.

**Project administration:** Scott C. Weaver, Ping Wu.

**Resources:** Pei-Yong Shi, Ping Wu.

**Supervision:** Scott C. Weaver, Ping Wu.

**Writing – original draft:** Pei Xu, Scott C. Weaver, Ping Wu.

**Writing – review & editing:** Pei Xu, Junling Gao, Chao Shan, Tiffany J. Dunn, Xuping Xie, Hongjie Xia, Jing Zou, Beatriz H. Thames, Amulya Sajja, Yongjia Yu, Alexander N. Freiberg, Nikos Vasilakis, Pei-Yong Shi, Scott C. Weaver, Ping Wu.

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
