## [Decision Letter · Decision Letter 0]

14 Dec 2020

Dear Dr. Wu,

Thank you very much for submitting your manuscript "Inhibition of innate immune response ameliorates 

Zika virus-induced neurogenesis deficit in human neural stem cells" for consideration at PLOS Neglected Tropical Diseases. As with all papers reviewed by the journal, your manuscript was reviewed by members of the editorial board and by several independent reviewers. The reviewers appreciated the attention to an important topic. Based on the reviews, we are likely to accept this manuscript for publication, providing that you modify the manuscript according to the review recommendations. 

Sincerely,

Gregory Gromowski

Associate Editor

Sunit Singh

Deputy Editor

Reviewer's Responses to Questions

**Key Review Criteria Required for Acceptance?**

**Methods**

-Are the objectives of the study clearly articulated with a clear testable hypothesis stated?

-Is the study design appropriate to address the stated objectives?

-Is the population clearly described and appropriate for the hypothesis being tested?

-Is the sample size sufficient to ensure adequate power to address the hypothesis being tested?

-Were correct statistical analysis used to support conclusions?

-Are there concerns about ethical or regulatory requirements being met?

Reviewer #1: (No Response)

Reviewer #2: -Yes

-Partially (comments bellow)

-Partially (comments bellow)

-Partially (comments bellow)

-Yes

-Yes

**Results**

-Does the analysis presented match the analysis plan?

-Are the results clearly and completely presented?

-Are the figures (Tables, Images) of sufficient quality for clarity?

Reviewer #1: (No Response)

Reviewer #2: -Yes

-Yes, but see comments to authors

-Yes

**Conclusions**

-Are the conclusions supported by the data presented?

-Are the limitations of analysis clearly described?

-Do the authors discuss how these data can be helpful to advance our understanding of the topic under study?

-Is public health relevance addressed?

Reviewer #1: (No Response)

Reviewer #2: -Partially

-Yes

-Yes

-Yes

**Editorial and Data Presentation Modifications?**

Reviewer #1: (No Response)

Reviewer #2: (No Response)

**Summary and General Comments**

Reviewer #1: Zika virus (ZIKV) and its strong link to microcephaly have raised major public health concerns. Although ZIKV infection induces strong innate immune responses and the infection has been reported to affect the proliferation and differentiation of neural stem/progenitor cells (NS/PCs) both in vitro and in animal models. However, it is unclear whether and how innate immune response affects neurogenesis. In this study, the authors used an Asian-American lineage ZIKV strain to infect primary human NS/PCs originally derived from fetal brains. They found that ZIKV strongly activated several key molecules in the innate immune pathways to impair neurogenesis in a cell stage-dependent manner. Inhibition of the overactivated innate immune responses with FAMP ameliorated ZIKV-induced defects in neurogenesis. 

This study has provided novel evidence that ZIKV infection induced innate immune response plays an important role in NS/PC development. In addition, the study suggests that orchestrating the host innate immune responses in NS/PCs after ZIKV infection could be promising therapeutic approach to attenuate ZIKV-associated neuropathology. I have several questions for the authors to answer.

1. Blocking TNF alpha with antibody has been shown to have protective role in mouse model to block cell death and increase survival. Does FAMP block cell death in NS/PCs? 

2. “Among those 2% of ZIKV-infected cells, nearly 90% were GFAP+ astrocytes, which were 8 times more abundant than Tuj1+ neurons”. Does this mean that ZIKV infects astrocyte more efficiently than neuron or there are more astrocytes in the culture? Or, ZIKV infection leads to the differentiation of NS/PCs to astrocyte other than neuron?

3. In addition to the innate immune response factors tested in this study, several other pathways components have also been reported. The authors should discuss why this study focuses on STAT1 while ignored the others, for example, IL6 and TNF alpha etc.?

Reviewer #2: In Xu et al.'s manuscript, authors used two cell lines of primary hNSC derived from fetuses and the Puerto Rico ZIKV strain and found that ZIKV infection decreases neurogenesis, mediated by an exacerbated innate immune response. Additionally, the authors suggested a mechanism related to STA1 and neurogenesis impairment, based on FAMP treatment, which could be considered a therapeutic strategy. 

The manuscript brings novelty and presents more information concerning ZIKV infection in the CNS. Some points I would like to discuss: 

• Authors do not measure neuronal death, either by apoptosis, autophagy, or any other cell mechanism, which seems to be relevant in ZIKV infection during the neurogenesis process, in vitro or in vivo (Cugola et al., nature, 2016). 

• Neurogenesis detrimental after ZIKV infection in neuro progenitor also pointed out other processes that should be discussed here (Rosa Fernandes et al., Frontiers in Cellular Neuroscience, 2019). 

• NSC differentiation protocol used here differentiates both neurons and astrocytes?

• How many pics were used to quantify fig 2C? How many biological replicates were done? 

• Figure 3: What is the percentage of neurons and astrocytes in the differentiation protocol in control and ZIKV infected? 

• Fig 4F is written "poliferating"

• Authors suggested that the severity of neurogenesis impairment in hNS/PCs is caused by the extent of innate immune activation in the host cells. It would be essential to measure some cytokines and chemokines releasing in the cell culture system, not only by RT-qPCR. So, to associate the neurogenesis impairment of ZIKV infection only focusing on innate immune response, mainly based on FAMP inhibition of STAT1, is very interesting but sounds incomplete. The neurogenesis impairment is just based on the neuronal (TUJ1+ cells) counting?

PLOS authors have the option to publish the peer review history of their article (what does this mean?). If published, this will include your full peer review and any attached files.

Reviewer #1: Yes: Zhiheng Xu

Reviewer #2: No
---

## [Decision Letter · Decision Letter 1]

26 Jan 2021

Dear Dr. Wu,

We are pleased to inform you that your manuscript 'Inhibition of innate immune response ameliorates

Zika virus-induced neurogenesis deficit in human neural stem cells' has been provisionally accepted for publication in PLOS Neglected Tropical Diseases.

Best regards,

Gregory Gromowski

Associate Editor

Sunit Singh

Deputy Editor

Reviewer's Responses to Questions

**Key Review Criteria Required for Acceptance?**

**Methods**

-Are the objectives of the study clearly articulated with a clear testable hypothesis stated?

-Is the study design appropriate to address the stated objectives?

-Is the population clearly described and appropriate for the hypothesis being tested?

-Is the sample size sufficient to ensure adequate power to address the hypothesis being tested?

-Were correct statistical analysis used to support conclusions?

-Are there concerns about ethical or regulatory requirements being met?

Reviewer #1: yes

**Results**

-Does the analysis presented match the analysis plan?

-Are the results clearly and completely presented?

-Are the figures (Tables, Images) of sufficient quality for clarity?

Reviewer #1: yes

**Conclusions**

-Are the conclusions supported by the data presented?

-Are the limitations of analysis clearly described?

-Do the authors discuss how these data can be helpful to advance our understanding of the topic under study?

-Is public health relevance addressed?

Reviewer #1: yes

**Editorial and Data Presentation Modifications?**

Reviewer #1: (No Response)

**Summary and General Comments**

Reviewer #1: (No Response)

PLOS authors have the option to publish the peer review history of their article (what does this mean?). If published, this will include your full peer review and any attached files.

Reviewer #1: **Yes: **Zhiheng Xu

---

## [Editor Report · Acceptance letter]

23 Feb 2021

Dear Dr. Wu,

We are delighted to inform you that your manuscript, "Inhibition of innate immune response ameliorates
Zika virus-induced neurogenesis deficit in human neural stem cells," has been formally accepted for publication in PLOS Neglected Tropical Diseases.

Best regards,

Shaden Kamhawi

co-Editor-in-Chief

Paul Brindley

co-Editor-in-Chief
